# A Comparative Assessment of Agronomic and Baking Qualities of Modern/Old Varieties and Landraces of Wheat Grown in Calabria (Italy)

**DOI:** 10.3390/foods11152359

**Published:** 2022-08-06

**Authors:** Giovanni Preiti, Antonio Calvi, Angelo Maria Giuffrè, Giuseppe Badagliacca, Nino Virzì, Monica Bacchi

**Affiliations:** 1Department of AGRARIA, University Mediterranea of Reggio Calabria, 89122 Reggio Calabria, Italy; 2CREA–Council for Agricultural Research and Economics, Research Centre for Cereal and Industrial Crops, 95024 Acireale, Italy

**Keywords:** wheat, *Triticum aestivum*, *Triticum turgidum*, landraces, gluten index, baking quality

## Abstract

The cultivation of wheat has been part of the evolution of human civilisation since ancient times. Wheat breeding has modified some of its characteristics to obtain improved varieties with high production potential that better meet the demands of the bread and pasta industry. Even today, there are still old varieties, landraces, adapted to particular environments. They are still cultivated in some areas because of the interest shown by the market in typical bakery products expressing the cultural heritage of local communities. The aim of this work was to evaluate the bio-agronomic and bakery characteristics of four modern genotypes, one old cultivar and two landraces of wheat typically grown in Calabria (Southern Italy). The experiment was carried out over two years in two different locations, during which the main bio-agronomic and quality traits related to bread making aptitude were detected. A marked difference was found between the landraces and the other genotypes in both agronomic and technological characteristics. Despite the higher protein and gluten content, landraces were found to have a significantly lower gluten index.

## 1. Introduction

Wheat represents a staple crop in temperate areas since prehistoric times, providing a basic resource of nourishment for the human population and as cattle feed. Well-known for its high nutritional qualities, due to its content in starch, nitrogen compounds, micronutrients and dietary fiber, but also for its adverse effects on sensitive individuals suffering from celiac disease or allergies, it provides the basis for the production of diversified foodstuffs [1]. Global wheat production in 2020 was estimated at roughly 761 million tons from an area of approximately 219 million hectares [2].

The two most widely cultivated wheat species in the world are: common wheat (*Triticum aestivum* L.) also known as “bread” or “soft”, and durum wheat (*Triticum turgidum* L. var. *durum* Desf.) also referred to as “pasta” or “hard” wheat [3]. The former accounts for 95% of global wheat production [4]. With a variety of products more diverse than any other commodity crop, food made from wheat can provide up to 20% of human energy requirements [5]. Although durum wheat is also used as an ingredient in typical breads, particularly in some areas of Southern Italy, it represents the main cereal for pasta production [6,7]. Common wheat is used in a wide variety of food and beverage applications, from the production of bread, pasta and oriental noodles [8] to malting and brewing beer [9]. Since its domestication, it has been subjected to a range of breeding techniques in order to improve certain traits useful for human purposes [10]. Improvements in terms of yield and breadmaking quality were mainly achieved through a conventional farming management based on high agronomic inputs with increased use of material and energy resources and, nevertheless, pollution-related concerns [11]. Indeed, contemporary society has to cope with a variety of issues related to environment, population growth, food security and safety. It is well known that modern wheat varieties possess better agronomic and baking characteristics than older cultivars as a result of genetic improvement. The latter may have a higher mineral content, depending on the growing area, and for whose flour consumers are more interested and accept higher prices [12]. Landraces were maintained by farmers through the decades. They were the most widespread varieties in the northwest of the Old Continent until before the Green Revolution, adapting themselves to certain cultivation environments [13]. As reported in Zeven [14], the term landrace designates a variety characterized by high yield stability and tolerance to biotic and abiotic stress conditions, capable of giving intermediate production yields under low agronomic input. Some outcomes of the transition from landraces to improved varieties have been a decrease in the protein percentage of the caryopsis [15] along with an increased gluten content [16] and genetic homogeneity among genotypes [17]. Significant differences are reported comparing the agronomic characteristics between old and modern varieties, the former presenting generally a lower yield [18], higher plant height and hectoliter weight [19] and increased 1000 kernel weight [20]. Variability occurs as well with regard to the quality characteristics of flours derived from old wheat varieties. As reported in Boukid et al. [21], bread made from the old wheat variety Abbondanza showed the highest specific volume while achieving good consumer acceptance. The ability of old wheat populations to give consistent or even higher yields than commercial varieties, under low input and in marginal areas, could be an initial response to some matters mentioned above [22], as a genetic resource and for the preservation of biodiversity [23]. Finally, they would provide a valuable food production resource for farmers and traders targeting niche markets [24].

The present work involved the evaluation of the agronomic traits of old and modern wheat genotypes together with the characterization of the rheological and technological properties of their respective flours and breads, with the aim of enhancing local genotypes for the production of traditional baked products.

## 2. Materials and Methods

### 2.1. Site Characteristics, Experimental Design and Raw Materials

Two field experiments were conducted during the growing seasons 2014/15 and 2015/16 in two different locations in the countryside of the municipalities of Rombiolo (38°36′ N to 15°58′ E; m 568 a.s.l.) and Maierato (38°42′ N to 16°10′ E; m 362 a.s.l.) in Vibo Valentia province, Calabria (Italy). The first location is comprised within the perimeter of the Monte Poro Plateau, the main cereal-growing area of the province. The location of the sites that hosted the experiments are shown in Figure 1.

The study areas are characterized by a typical Mediterranean climate with a rainy season from October to March and a dry summer during which occasional thunderstorms may occur. The localities fell within a thermo-climatic zone between 14 °C and 15 °C of mean annual isotherms with annual precipitation of approximately 1000 mm [25]. The soils that hosted trials were characterized by a sandy-loam texture with almost no skeleton, rich in organic substance, with a sub-acid reaction in Rombiolo and sub-alkaline in Maierato, with a medium-high cation exchange capacity. The field experiment was based on a randomized block design with three replications. Raw materials included two wheat landraces (Rosia and Mazzancoio), one old wheat cultivar (Abbondanza) and four recently developed wheat varieties (Bologna, Altamira, Solehio, PR22R58). Rosia and Mazzancoio are two local landraces of high size, widely cultivated in the hinterland of Vibo Valentia, where they are appreciated for bread making purposes and valued for the large amount of straw destined to livestock use. Abbondanza is an old variety (registered as conservation variety) established in the 1950s, that has gradually been replaced in cultivation. The latter have established themselves since the 2000s, conventionally classified on the basis of their qualitative and technological characteristics, so as to indicate their prevalent use. An Italian method known as ISQ (Synthetic Index of Quality) is used to classify common wheat in five quality classes with different end uses [26]. Altamira, PR22R58 and Solehio (included in the national register of varieties in 2003, 2002 and 2008) are catalogued as ordinary “bread making wheat” (FP), while Bologna variety (entered in the variety register in 2002) is classified as “improver wheat” (FF) for its excellent technological properties (Table 1). As shown by the officially certified seed quantities (2016–2020 average), Bologna is still the most widely cultivated soft wheat cultivar in Italy [27]. All the genotypes belong to the *Triticum aestivum* L. species, with the exception of Mazzancoio, a peculiar species of wheat (*Triticum turgidum* L. spp.) characterized by a floury endosperm.

Sowing was carried out in 14.4 m^2^ plots (1.44 × 10 m) with 18 cm spacing between rows, using a plot seeder (Vignoli) between the second and third decade of November. Fertilisation was carried out with 92 kg ha^−1^ of P_2_O_5_ and 36 kg ha^−1^ of N at the same time as sowing, and 50 units ha^−1^ of N in urea form were applied during coverage (stages 25–29 [29]). A mixture of pinoxaden, clopyralid, florasulam (Axial 60) and fluroxypyr meptil (Columbus) was applied to control weed growth. Grains were harvested at maturity (stages 93–97) with a “Wintersteiger” plot combine. The main bio-agronomic traits of the tested genotypes and quality traits on harvested grains and their respective flours and breads were evaluated.

### 2.2. Field Measurements

A number of bio-agronomic parameters were measured during the growing season: the plant heading period (HP), expressed in number of days from the first of April, and the plant height (PH) (stages 83–87) were measured close to harvest.

### 2.3. Quality Assessment and Grain Milling

After harvest, grains for each plot were weighted, and the yield (GY), reported in t ha^−1^, was determined and converted into a standard moisture of 13%. Hectolitre weight (HW) and hardness (H) were determined for each sample using an Infratec 1241 Grain Analyzer (FOSS Analytical A/S, Hillerød, Denmark) following the manufacturer’s guidelines. Thousand kernel weight (TKW) was determined with a seed counter (Sadkiewicz Instruments, Polonia) by counting the number of seeds contained in 25 g sample and relating to 1000 seeds by means of the proportion. The grain nitrogen content (Kjeldahl method) as well as the protein content (GP) were determined using the conversion factor of 6.25 [30].

### 2.4. Flour Quality Evaluation

A 10 kg grain sample for each genotype, year and location (pooled sample of the 3 replications) was conditioned for 24 h at 20 °C, until a 14% moisture content of the grains was reached, using an experimental Bona cylinder mill (Bona Labormill 4RB, Monza, Italy) and separating fine particles (<250 µm), with an extraction rate of about 55–60%. This percentage was considered optimal to perform the quality analyses, due to a larger grain particle size and a lower degree of damaged starch. Quality analyses of the grain, dough and bread were performed in duplicate.

The protein content of the flour (FP) was determined using an Infratec 1241 Grain Analyzer. Wet (WGC), and dry gluten (DGC) and gluten index (GI) were determined using a Glutomatic System (Glutomatic 2200, Centrifuge 2015, Glutork 2020; Perten Instruments AB, Huddinge, Sweden) according to the AACC method 38–11 [31]. The color parameter *b**, referred as yellow index (YI), was determined by Chroma meter CR-300 (Minolta, Osaka, Japan), under the illuminant, D65.

### 2.5. Dough Rheological Measurements

Alveograph parameters, i.e., deformation energy W (DEW) and alveograph ratio P/L (ARPL), were determined using an alveograph equipped with the software Alveolink NG (Chopin MA 87 model, Tripette and Renaud, Villeneuve-la-Garenne, France), according to the AACC method 54–30A [31]. Water absorption of the flour (WAC) at a maximum consistency of 500 FU (Farinograph Units), as well as the development time of the dough (FDDT) and dough stability (FS), degree of softening (FSD) and farinographic quality number (FQN), were determined using a Farinograph-E (Brabender, Germany) according to AACC 54–21 [30,31].

### 2.6. Bread-Making and Bread Quality Evaluation

The baking procedure was carried out according to the AACC method 10.10 [31]. In the experimental baking laboratory (temperature, 25 ± 2 °C), each loaf of bread was obtained adding 100 g of flour (14% moisture basis), compressed yeast (3%), sugar (6%), NaCl (2%), ascorbic acid (80 p.p.m) and shortening (3%) to distilled water. The amount of water required has been calculated as the difference between the water absorption, determined by farinographic analysis, and the water added with the solutions. The doughs were leavened at 29 ± 1.41 °C and at 82.5% ± 3.54% relative humidity in a thermostatic proofing cabinet equipped with a steam humidifier for 1.75 h. The dough was sheeted in a roller, then it was proofed for another 50 min, and a second sheeting roll was performed. The doughs were sheeted through the sheeting rolls, and after an additional 25 min, they were manually rounded and placed into individual metal bread moulds. The doughs were then proofed for another 50 min, for a total fermentation time of 3.83 h. They were then baked in a humidified, ventilated and thermostated electric oven for 18 min at 215–220 °C. Loaf bread volume (LBV) was determined in a loaf volume metre, according to the AACC method 10.05.01 [31]. Bread height (BH) was measured by using a digital calliper (Digi-MaxTM, Scienceware, N.J., USA). Bread weight (BW) was also assessed. The CIELAB space *L**, *a**, *b** colour parameters were measured for bread crumb and crust. The corresponding results are reported in the Appendix A. Crumb porosity (PO) was expressed based on the Mohs scale, ranging from 1 to 8, for higher and lower porosity, respectively. Crust texture (CT) was indicated on a scale varying from 1 to 4 (smooth to rough, respectively).

### 2.7. Statistical Analysis

Data analysis was realised using R programming language and environment [32], software version 4.1.0 (18 May 2021). The full list of R packages utilised [32,33,34,35,36,37,38,39,40,41] is available in the Appendix A. The Bartlett test indicated homogeneity of variance between years, and therefore, the ‘year’ factor has not been considered as a source of variation; consequently, a two-way analysis of variance with interaction (location and genotype) was performed. The statistical significance of the effects was analysed using F-tests, whereas the differences between means were tested using the Tukey’s HSD Test at a *p* ≤ 0.05 significance. In order to investigate the relationship between each pair of variables, correlation coefficients were estimated based on Pearson’s correlation method. Holm’s method was used for *p*-values adjustment for multiple comparisons. R package ggstatsplot [35] was used to compute the correlation between variables and to graphically display the correlation matrix. Principal component analysis was computed to reduce the dimensionality of the dataset so as to make it more interpretable. Variables were scaled before computing PCA to allow comparison between variables expressed by different units of measure. The crumb and crust data were not considered to perform well in estimating the source of variability and were not included in the PCA analysis. The first three principal components explained a variance of 83.3% (Appendix A). Only the first two components were used in this study to ensure a more effective representation and interpretation of the data. R package FactoMineR [36] was used to compute PCA and the package factoextra [37] to extract and visualize graphically PCA results.

## 3. Results

### 3.1. Bio-Agronomic and Grain Quality Traits

Modern varieties were characterized by high yields, ranging from 4.98 to 5.82 t ha^−1^, while the other genotypes presented lower values (<4 t ha^−1^), with Rosia resulting the least productive (2.93 t ha^−1^). The genotype–environment interaction term resulted significant (data not shown). In this regard, it is interesting to note that two modern varieties (Altamira and Bologna) showed significantly different GY between the two locations. On the contrary, GY remained almost identical for the old variety and the landraces between the two experimental areas (Table 2).

A considerable difference was found for the plant height between the different genotypes, as expected. The modern varieties presented shorter culms, ranging from 70 to 81 cm for PR22R58 and Altamira, respectively, while Abbondanza showed an intermediate PH (103 cm). Landraces were the tallest, with a maximum PH of 180 cm for Mazzancoio. The heading period varied from 32 days for Altamira to 49 days for Rosia. Modern genotypes therefore presented shorter HD, while no statistically significant difference has been found for Abbondanza and Mazzancoio. Only the genotype exerted a significant effect on the HP. All the genotypes showed ideal values for the hectolitre weight (≥76 kg hL^−1^) with the exception of Abbondanza. In particular, Mazzancoio presented the highest HW among all genotypes under study (Table 2). Thousand kernel weight was influenced by both genotype and location, as well as their interaction (data not shown). A clear discrepancy was found between Abbondanza and Mazzancoio (32.8 and 61.7 g, respectively). TKW was found to vary between modern genotypes, with a maximum value of 47.1 g for Altamira. Landraces were characterized by higher hardness values (65% for Rosia and 75% for Mazzancoio), while variability was detected among modern varieties, with values ranging from 17 to 61% for Solehio and Bologna, respectively. Grain protein content ranged from 10.4 for Solehio to 15.1% d.m. for Rosia. GP was significantly affected by both genotype and location. Landraces were characterized by the highest GP (Table 2), while PR22R58 and Solehio values were below 11% (d.m.).

### 3.2. Flour and Dough Quality Evaluation

As well as for the GP, Rosia presented the highest flour protein content (14.5% d.m.), while Solehio showed the lowest one (9.0% d.m.). Location and genotype, as well as their interaction (data not shown), exerted a strong influence on this quality trait. The yellow index ranged from 5.36 for Abbondanza to 11.89 *b** for Mazzancoio. Landraces were characterized by the highest values for the wet and dry gluten content but lower gluten indexes (34 and 8 for Rosia and Mazzancoio, respectively, as reported in Table 3). On the contrary, the other genotypes presented an inverse trend, with WGC ranging from 12.0 to 22.5%, DGC between 3.9 and 8.2% and higher GI. In particular, modern genotypes presented GI values close to 100%.

Landraces were characterized by lower values of DEW when compared with the old and modern genotypes. No significant difference was recorded for Abbondanza, PR22R58 and Altamira. The highest value, as expected, was recorded for Bologna (Table 4).

The modern varieties were characterized by similar ARPL, ranging from 1.13 for PR22R58 to 1.46 for Bologna. Mazzancoio showed a significantly different result (2.30) compared to all the other genotypes. On the contrary, Abbondanza and Rosia did not significantly differentiate, presenting the lowest ARPL (Table 4). The farinograph development time ranged from 1.53 to 2.35 min for PR22R58 and Rosia, respectively. As well as for the ARPL, Rosia and Abbondanza presented similar outcomes (2.24 min on average). FDDT of less than 2 min were recorded for all other varieties. The farinograph stability ranged from 0.54 min for Mazzancoio to 14.80 min for Bologna, a distinctive high value associated with a strong dough. All the other varieties presented no significant difference, with the exception of Abbondanza (Table 4). The farinograph softening degree varied in the range 54–177 BU for Bologna and Mazzancoio, respectively. Overall, landraces showed higher FSD (Table 4). Moreover, there was a significant difference among the modern varieties, with Solehio, Altamira and PR22R58 forming a different group from Bologna. The old variety presented a low FSD as well (Table 4). The farinograph water absorption ranged from 48.4 for Abbondanza to 66.5% for Rosia. Modern genotypes were characterized by intermediate WAC values, in the range 50.3–55.6% for Solehio and Bologna, respectively. The farinograph quality number ranged from 20.7 for Mazzancoio to 47.8 for Bologna. Location, variety and their interaction (data not shown) exerted a significant effect on FQN, overall.

### 3.3. Bread Quality Evaluation

The bread made from Abbondanza flour resulted in being the most voluminous (485.0 cm^3^). Solehio, PR22R58 and Mazzancoio did not exhibit any significant difference. In particular, Solehio and Mazzanocio had the lowest LBV (Table 5). The LBV of the landrace Rosia was comparable to that of some modern genotypes (PR22R58, Bologna and Altamira).

Bread height ranged from 58.1 for Mazzancoio to 88.3 mm for Abbondanza. No significant difference was found in terms of BH comparing Rosia with modern genotypes. The bread made from Mazzancoio was the heaviest (150.8 g), while the one baked with Abbondanza flour resulted in being the lightest (130.9 g). No significant difference was reported for the modern genotypes. Rosia presented a BW not statistically different from Bologna (Table 5). Crumb porosity resulted the same for the old genotype and the landraces (Table 5). The modern genotypes presented a value equal to 7, with the exception of Bologna, which presented the same value as the old and landraces genotypes. The crumb colour characteristics are reported in Appendix A. The *L** coordinate ranged from 69.4 to 75.9 for Mazzancoio and Abbondanza, respectively. All the genotypes presented negative values for the *a** coordinate, except for Mazzancoio, that also presented the highest value (18.5) for the *b** coordinate. Bread crust texture was expressed with a scale ranging from 1 to 4. The modern genotypes showed the lowest value, while the old genotype and the landraces were characterized by a value equal to 2. The crust colour characteristics are reported in the Appendix A. *L** coordinate for the crust ranged between 36.6 for Mazzancoio to 47.8 for Abbondanza (Appendix A), showing the same trend for the same qualitative trait referred to the crumb. Mazzancoio and Rosia presented the lowest value for the *a** coordinate when compared to all the other genotypes (Appendix A). The *b** coordinate for the crust was higher for the modern genotypes, while the landraces presented lower values. Abbondanza had the greatest and significantly different outcome (Appendix A).

## 4. Discussion

A marked difference in yield was observed in this study, in accordance with the results reported by Frankin et al. [42]. It is well known that technological, biological and environmental factors affect this important crop trait [43]. A historic contribution to the increase in wheat yield can be attributed to the Italian agronomist and plant breeder Nazareno Strampelli, whose research work during the “Battle of Wheat” (after 1925) led to doubling of national yields over a decade [44]. That approach aimed at enhancing the crop performance in high-input systems has continued over time with the gradual replacement of old, low-yielding varieties with improved ones [45]. In this study, the modern genotypes resulted in being more productive than the old one and the landraces. The genetic difference and the pedo-climatic conditions exerted a significant influence on some modern varieties (Altamira and Bologna). Landraces did not exhibit significant fluctuations between years and different locations, confirming what is reported in Zeven [14] with reference to their production stability. Along with increased yields, another consequence of the “Green Revolution” was the introduction of dwarfing traits with the aim of reduce plant height and, consequently, the risk of lodging [46]. As expected, indeed, modern genotypes presented shorter culms than the old variety and the landraces. A similar trend was observed with regard to the heading period, which was generally faster for modern genotypes. This represents a further consequence of the breeding improvement on wheat, which allows improved varieties to avoid dry periods in the late season when caryopsis filling occurs [44].

Grain quality is crucial in terms of flour yield at the milling stage. Nevertheless, it influences the final quality of the derived bakery products. For this reason, the hectolitre weight, thousand kernel weight, hardness and protein content were determined as well. The hectolitre weight is defined as the ratio between the mass of a cereal sample and the volume occupied when placed in a container under defined conditions [47]. Stress factors during cereal growth and genetic differences affect this physical quality trait. Measurements above 76 kg hL^−1^ are associated with high quality wheat for the milling industry [48]. Abbondanza was the only genotype showing a HW below the optimal threshold mentioned above, while some variability was found between modern genotypes and landraces. Thousand kernel weight is an indicator of seed weight and size related to yield and milling quality traits [49]. There has been a gradual increase from the old variety to the landraces, with intermediate values for the modern genotypes, as also reported by Ruiz et al. [50]. Hardness is related to kernel texture, and it is used to distinguish wheat into categories such as “hard” or “soft” [51]. As expected, modern genotypes presented lower H values, which reflected their suitability to obtain flours for bread production. Taking into consideration the variability of environmental and genetic factors, a difference of 1–1.5% in protein content is common between landraces and modern varieties [45]. With the exception of Bologna, which presented the highest GP among the modern genotypes, a statistically significant difference was detected between the latter and the landraces. Rosia had the highest GP, which was found to be constant among years and locations. As confirmed by correlation analysis and consequently depicted in the correlogram (Figure 2), GY showed an opposite trend to that of HP (r: −0.84), PH (r: −0.73) and GP (r: −0.53), in accordance with Iqbal et al. [52].

The resulting biplot from the PCA (Figure 3) clearly shows the negative correlation between the variables located in the opposite quadrants from the origin of the graph. This confirms that higher GY, shorter HP and lower PH are typical attributes of modern varieties.

Furthermore, a negative but not statistically significant correlation was found between GY, HW and TKW (Figure 2), in contrast to the findings of Assefa [53], in which a positive correlation has been reported. The two landraces formed two distinct groups in the opposite direction of GY, with Mazzancoio individuals (observations) located on the same side of HW and TKW (Figure 3), thus presenting high values for these variables. As depicted in Figure 2, H resulted strongly correlated with GP (r: 0.79), WGC (r: 0.83) and DGC (r: 0.81).

Flours obtained by landraces were characterized by high FP (Table 3). Except for Solehio, all genotypes exceeded the threshold of 9.0% under the relevant Italian law, DPR 09/02/2001 n. 187 [54] for ‘type 00’ wheat flour. Carotenoids and anthocyanins are the main pigments that determine the colour of flour and semolina. Their presence is strongly requested in bakery products thanks to their nutritional, technological and hedonistic benefits [55]. The quantity of intrinsic pigments in the flour influences the measurement of the yellow index. Flours with low *b** are suitable and preferred for white bread production [56]. From all the genotypes under study, it was possible to produce flours with an average YI of less than 10 *b**. The high and significantly different value for Mazzancoio could be attributed to the fact that it belongs to *Triticum turgidum* L. spp., although its caryopsis has a peculiar floury fracture; for such features, it was used in the past by local communities to make a traditional bread mixed with other soft wheat flours. The quantity and quality of gluten were significantly affected by genetic differences among genotypes. Indeed, genetic improvement has also been directed towards enhancing technological characteristics for the bakery industry. Consequently, there has been an increase in the gluten index from landraces to modern varieties [45]. Gluten is defined as a visco-elastic protein mass obtained during the kneading of certain flours (made of wheat, barley, rye, oats and their crossbred varieties) and water, which gives the flour a high water-absorption capacity and the ability to retain the gas formed during leavening [57]. Gluten strength is a complex trait influenced by various quali-quantitative properties of gluten, such as the type and ratio of glutenin and gliadin subunits. The consequences of genetic improvement have also resulted in an increase in gluten strength from landraces to modern varieties. The former are generally characterised by GI values ranging from 6 to 32%; the latter by higher levels varying from 55 to 87% [45]. Indeed, a clear difference between the genotypes was detected regarding quantity and quality of gluten (Table 3). The debate on the forms of disease related to gluten quantity and quality is still open; however, a protein fragment related to coeliac disease appears to be more frequent in modern than in old wheat genotypes and landraces [58,59].

A higher gluten content was quantified in landraces, which, however, corresponded to a medium or poor quality by reason of low GI (8–34%). On the contrary, the other genotypes showed a completely opposite trend, with exceptionally higher GI for modern genotypes (70–100%). Among them, Bologna presented the highest WGC with a value similar to that reported in Bosi et al. [60]. WGC and DGC were highly negatively correlated with GI (r: −0.89, −0.82, respectively). Nevertheless, WGC was positively correlated to GP (r: 0.82), in accordance with the findings of Bonfil and Posner [61]. Furthermore, a negative and significant correlation between GI and GP was detected (Figure 2), contrary to the results reported by Bonfil and Posner [61], where a weak positive correlation has been reported. GI and GP presented completely divergent orientations on the PCA biplot (Figure 3). In addition, they contributed most to the construction of the first two PCs (Appendix A). Based on specific gluten index ranges [62], modern genotypes were characterized by a strong gluten (GI > 80%) of excellent quality. Rosia and Abbondanza presented a normal gluten (30% < GI > 80%). On the other hand, Mazzancoio was characterized by a weak gluten of poor quality (GI < 30%). If compared to GI values reported in another study on durum wheat landraces, Mazzancoio recorded a lower result overall [63]. The deformation energy W represents the energy required to inflate a dough sample until it breaks [64]. In this study, DEW was influenced by both variety and location. Moving from landraces to modern genotypes, there has been an increase in DEW, with the maximum value recorded for Bologna. In particular, Mazzancoio was characterized by the lowest value among genotypes, while Rosia and Solehio showed a value not significantly different. The sole use of flour obtained from Mazzancoio grains would then not provide an ideal glutinic network. It would therefore be preferable to use it in blends with flours with better baking features. DEW was negatively strongly correlated with FSD (r: −0.89), while no significant correlation was detected with traits related to GP, FP and gluten content (related vectors on the PCA biplot form approximately an angle of 90°, Figure 3). A positive correlation between DEW and GI (r: 0.59) was also found, in accordance with Bornhofen et al. [64]. The modern genotypes under study are generally characterized by high DEW, such as to receive the denomination of improver wheat (Bologna) or ordinary bread making wheat (Altamira, Solehio, PR22R58). In the present study, they showed lower DEW, overall. This variability could be attributed to the particular pedoclimatic conditions of the study areas. Indeed, the effects of year and locality, as well as their interaction, play a preponderant role in defining the agronomic and qualitative characteristics of wheat [65]. The ratio between tenacity and extensibility of dough (P/L), also known as configuration ratio, is commonly used together with the deformation energy as a quality flour indicator [64]. Modern genotypes showed similar ARPL values. Mazzancoio showed the highest value, typical of a dough that is hard to process. On the contrary, Abbondanza and Rosia showed a low ARPL, indicative of an extensible and weak dough. The farinograph development time represents the kneading interval necessary to obtain the optimal development of a dough [47]. With exception of Rosia and Abbondanza, all genotypes presented FDDT less than 2 min. High values make it possible to obtain dough that can withstand long kneading and rising times. FDDT was positively correlated with GP (r: 0.49). The time interval during which a dough remains at its maximum consistency is known as farinograph stability [47]. Most genotypes were characterized by very low FS, in particular Mazzancoio, which maintained maximum consistency for an average time of less than 1 min. Bologna presented the highest value, significantly different from all the other genotypes. FS resulted in being highly and positively correlated with DEW (r: 0.80). The farinograph softening degree represents the decrease in the dough consistency after a set interval compared to the standard consistency of 500 BU (Brabender Unit) [47]. High values were associated with landraces, while lower ones have characterized the old and the modern genotypes. The high FS and low FSD shown at the same time by Bologna confirmed that this genotype presents excellent breadmaking aptitude. FSD was negatively correlated with GI (r: −0.76), DEW (r: −0.89) and FS (r: −0.64). The farinograph water absorption is defined as the amount of water to add to flour to reach the optimal consistency of 500 BU [47]. Starch and gluten influence this parameter, with high values commonly preferred [66]. Mazzancoio required the highest amount of water to develop an optimal dough consistency, which combined with the highest FSD, indicated a flour of poor quality [67]. As specified in the PCA biplot, in fact, the vectors of these two variables are close to Mazzancoio individuals. Overall, WAC was positively correlated with GP (r: 0.56), YI (r: 0.69), WGC (r: 0.77), ARPL (r: 0.47) and FSD (r: 0.55) and negatively correlated with GI (r: −0.73). FQN was positively correlated with FDDT (r: 0.4) and FS (r: 0.63), but an inverse correlation was found in relation to FSD (r: −0.56), as also reported in Jańczak-Pieniążek et al. [68], where a stronger negative correlation was found. Breads made from the flour obtained from the two landraces were characterized by a significant difference. Moreover, Rosia presented similar values when compared with the modern genotypes under study. Unexpectedly, Solehio presented the lowest volume, as well as Mazzancoio (Table 5). LBV was negatively correlated with ARPL (r: −0.43) and FSD (r: −0.46), while a positive correlation with DEW (r: 0.46) and FQN (r: 0.56) was found. BH was positively correlated with GI (r: 0.51), DEW (r: 0.59), FQN (0.52) and LBV (0.77), while a negative correlation with ARPL (r: −0.62), FSD (r: −0.71) and WAC (r: −0.67) was detected. Breads obtained from the landraces were the heaviest. There was a strong correlation between BW and WAC (r: 0.92), while an inverse relationship was found with BH (r: −0.67) and GI (r: −0.62). There was no difference for the crumb porosity between the old genotype and the two landraces, while the modern genotypes presented a higher value overall. No statical difference has been detected for the *L**Rosia and Mazzancoio presented a similar value. In particular, Mazzancoio had the lowest value for the *L** coordinate, but the greater for the *a** and *b** coordinates, thus presenting a more pronounced yellowish colour than the common wheat genotypes, as expected. The crust of the breads obtained from modern genotypes were characterised by a softer texture than the old genotype and landraces. Statistically significant differences were found for all colour parameters measured on the crust between the three genotype categories under study. Mazzancoio generally presented a more yellowish colour even in this case, with intermediate yellow index for Abbondanza and Rosia.

## 5. Conclusions

In this study, the quality traits of grains, flours, doughs and breads obtained from different wheat genotypes were evaluated. The agronomic study on selected genotypes confirmed the better performance of modern varieties compared to unimproved ones. The landraces, with lower yields, were characterised by a higher protein content and a low gluten index. The cultivation of old grains allows the use of less impactful technologies and the reduction in waiting times for raw materials in short supply chains, with consequent advantages in terms of quality and wholesomeness. Furthermore, the productive stability of these wheats in different environments and years, their rusticity and competitiveness against weeds is of particular importance in low-input and organic farming systems.

As was to be expected, with regard to the technological quality parameters, a clear discrepancy was found; the breadmaking properties of modern varieties were found to be better, with significant differences compared to old varieties. However, it would be interesting to study the best combination of flours from different genotypes. Flours from modern varieties and landraces could be used in different mixtures to obtain traditional bakery products that are very closely related to the cultivation area. The exploitation of the latter would also make it possible to keep in cultivation genotypes otherwise destined to disappear, contributing to the loss of genetic diversity.

## Figures and Tables

**Figure 1 foods-11-02359-f001:**
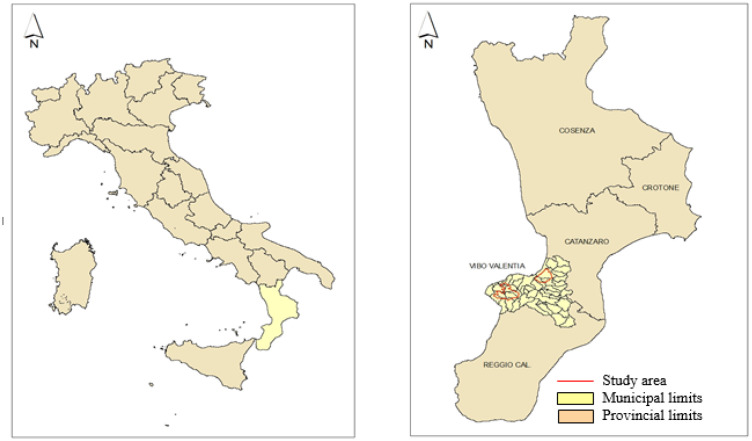
Study areas in the province of Vibo Valentia.

**Figure 2 foods-11-02359-f002:**
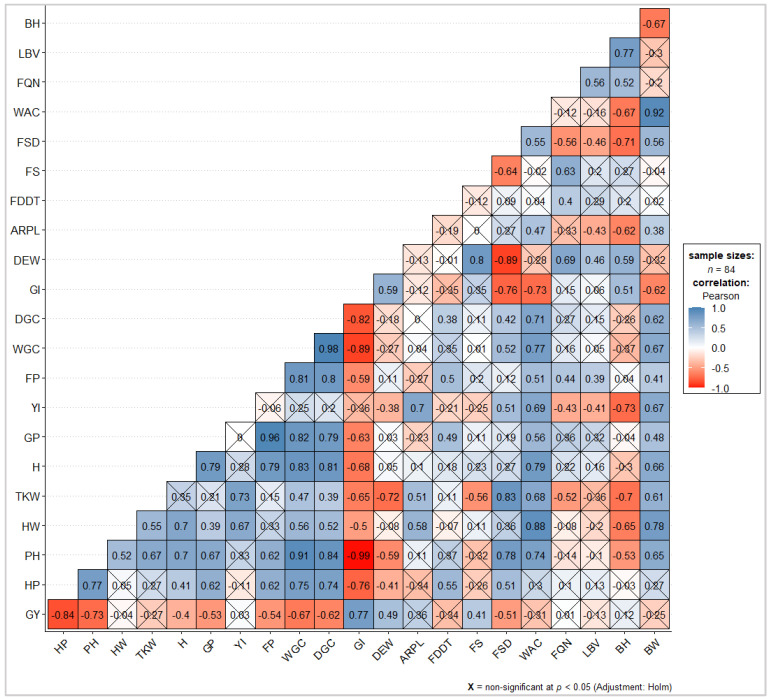
Correlation matrix of all pairs of variables. Correlation coefficients on the principal diagonal are not shown. Flagged coefficients are not significant (at 5% significance level). Negative and positive correlations are displayed in red and steelblue, respectively, while white colour represents no association between pairs of variables. Colour intensity is proportional to the correlation coefficients values, as reported in the legend. GY, grain yield; HP, heading period; PH, plant height; HW, hectolitre weight; TKW, thousand kernel weight; H, hardness; GP, grain protein; YI, yellow index; FP, flour protein; WGC, wet gluten content; DGC, dry gluten content; GI, gluten index; DEW, deformation energy W; ARPL, alveograph ratio P/L; FDDT, farinograph dough development time; FS, farinograph stability, FSD, farinograph softening degree; WAC, water absorption capacity; FQN, farinograph quality number; LBV, loaf bread volume; BH, bread height; BW, bread weight.

**Figure 3 foods-11-02359-f003:**
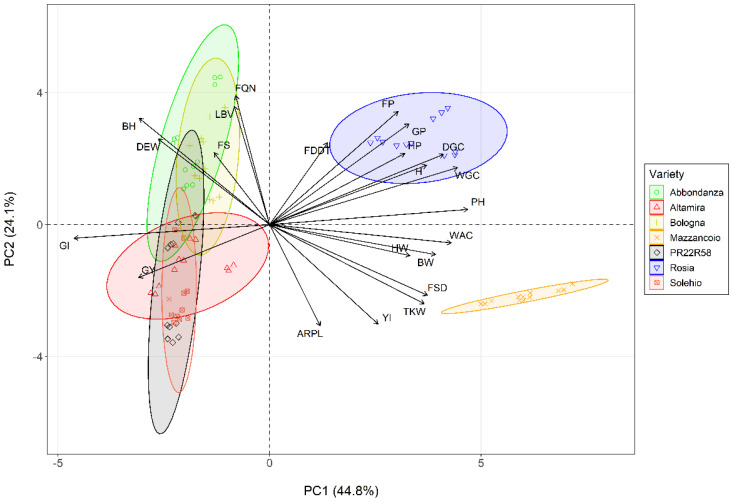
Biplot of principal component scores and loading vectors. Observations are shown as points with different shapes as described by the legend. Variables are shown as vectors. The first component (PC1) accounts for 44.8% of the variance, while the second component (PC2) for the 24.1%. Point concentration ellipses are drawn assuming a multivariate t-distribution with a confidence interval of 95%. GY, grain yield; HP, heading period; PH, plant height; HW, hectolitre weight; TKW, thousand kernel weight; H, hardness; GP, grain protein; YI, yellow index; FP, flour protein; WGC, wet gluten content; DGC, dry gluten content; GI, gluten index; DEW, deformation energy W; ARPL, alveograph ratio P/L; FDDT, farinograph dough development time; FS, farinograph stability, FSD, farinograph softening degree; WAC, water absorption capacity; FQN, farinograph quality number; LBV, loaf bread volume; BH, bread height; BW, bread weight.

**Table 1 foods-11-02359-t001:** Characteristics of the wheat selected for the experiment.

Genotype	ISQ	Genealogy	Maintenance Manager
Bologna	(FF)	(H89092 × H89136) × Soissons	356-ETS Claude Camille Benoist, 159-Venturoli Sementi S.R.L.
Altamira	(FP)	96,248 × Isengrain	1168-Nickerson International Research Geie, 1242-Limagrain Italia S.P.A
Solehio	(FP)	Isengrain × Ornicar	441-Kws Momont SAS
PR22R58	(FP)	(Victo × FVP0040) × XXC31	1057-Pioneer Genetique S.A.R.L., 681-Pioneer Hi-bred Int. Inc., 53-Pioneer Hi-Bred Italia servizi agronomici SRL
Abbondanza	NA	Autonomia x Fontarronco	1674-Molini Cicogni SRL, 1471-Arcoiris SRL
Rosia	NA	NA	Local farmers
Mazzancoio	NA	NA	Local farmers

ISQ: Synthetic Index of Quality; FF (*frumento di forza*): improver wheat; FP (*frumento panificabile*): ordinary bread making wheat; NA: not available. Maintenance manager information was obtained at [28].

**Table 2 foods-11-02359-t002:** Bio-agronomic and grain quality traits.

	GY	PH	HP	HW	TKW	H	GP
**Genotype**	***	***	***	***	***	***	***
Altamira	5.82 ± 0.62 ^a^	81 ± 4.26 ^d^	32 ± 2.35 ^e^	77.6 ± 2.25 ^c^	47.1 ± 1.70 ^b^	37 ± 15.94 ^c^	11.6 ± 1.05 ^d^
Bologna	5.65 ± 0.72 ^ab^	78 ± 3.41 ^d^	35 ± 1.70 ^cd^	80.2 ± 1.46 ^b^	34.3 ± 1.96 ^d^	61 ± 9.49 ^b^	12.7 ± 0.81 ^bc^
Solehio	5.17 ± 0.51 ^bc^	77 ± 2.47 ^d^	36 ± 2.01 ^c^	76.7 ± 2.28 ^cd^	44.4 ± 3.80 ^c^	17 ± 4.23 ^d^	10.4 ± 0.43 ^e^
PR22R58	4.98 ± 0.45 ^c^	70 ± 2.78 ^d^	34 ± 1.90 ^de^	76.6 ± 2.07 ^e^	42.8 ± 2.07 ^c^	34 ± 9.75 ^c^	10.5 ± 1.03 ^e^
Abbondanza	3.51 ± 0.33 ^d^	103 ± 4.61 ^c^	41 ± 1.41 ^b^	74.8 ± 0.74 ^d^	32.8 ± 0.69 ^d^	41 ± 5.51 ^c^	12.1 ± 0.76 ^cd^
Mazzancoio	3.57 ± 0.16 ^d^	180 ± 5.94 ^a^	41 ± 1.51 ^b^	84.5 ± 2.88 ^a^	61.7 ± 2.19 ^a^	75 ± 1.38 ^a^	13.2 ± 0.12 ^b^
Rosia	2.93 ± 0.16 ^e^	160 ± 6.50 ^b^	49 ± 1.61 ^a^	77.9 ± 1.13 ^bc^	48.1 ± 2.72 ^b^	65 ± 11.52 ^ab^	15.1 ± 1.52 ^a^
**Location**	**	**	ns	**	*	**	***
Rombiolo	4.40 ± 0.97 ^b^	106 ± 42.30 ^b^	38 ± 5.16 ^a^	78.2 ± 3.29 ^a^	45.0 ± 8.85 ^a^	50 ± 20.10 ^a^	12.6 ± 1.70 ^a^
Maierato	4.64 ± 1.34 ^a^	108 ± 41.95 ^a^	38 ± 6.06 ^a^	78.5 ± 3.74 ^a^	43.9 ± 9.76 ^b^	44 ± 22.33 ^b^	11.8 ± 1.76 ^b^
**Variety**							
Modern	5.41 ± 0.66	76 ± 5.22	34 ± 2.40	77.8 ± 2.47	42.1 ± 5.43	37 ± 19.21	11.3 ± 1.28
Old	3.51 ± 0.33	103 ± 4.61	41 ± 1.41	74.8 ± 0.74	32.8 ± 0.69	41 ± 5.51	12.1 ± 0.76
Landrace	3.25 ± 0.36	170 ± 12.08	45 ± 4.01	81.2 ± 3.99	54.9 ± 7.38	70 ± 9.44	14.1 ± 1.44

GY, grain yield (t ha^−1^); PH, plant height (cm); HD, heading period (d); HW, hectolitre weight (kg hL^−1^); TKW, thousand kernel weight (g); H, hardness (%); GP, grain protein (% d.m.). Mean values ± standard deviation are reported for each variable. Different letters in the same column indicate significant differences according to Tukey HSD test at *p* ≤ 0.05, while ***, **, *, ns, indicate the following levels of significance for each ANOVA model: 0.001, 0.01, 0.5, not significant. Only the mean values ± standard deviation are shown for the variety component for descriptive purposes. Sources of variability are indicated in bold in the first column of the table.

**Table 3 foods-11-02359-t003:** Flour quality characteristics.

	FP	YI	WGC	DGC	GI
**Genotype**	***	***	***	***	***
Altamira	10.7 ± 1.08 ^cd^	7.77 ± 0.77 ^d^	12.9 ± 1.36 ^d^	4.1 ± 0.42 ^d^	99 ± 1.38 ^a^
Bologna	12.5 ± 0.83 ^b^	7.79 ± 0.25 ^d^	22.5 ± 0.73 ^b^	8.2 ± 0.25 ^b^	99 ± 1.04 ^a^
Solehio	9.0 ± 0.93 ^e^	8.85 ± 0.40 ^c^	12.0 ± 0.64 ^d^	4.2 ± 0.46 ^d^	100 ± 0.67 ^a^
PR22R58	9.9 ± 1.37 ^de^	9.66 ± 0.71 ^b^	12.6 ± 1.75 ^d^	3.9 ± 0.70 ^c^	99 ± 1.34 ^a^
Abbondanza	11.6 ± 0.78 ^bc^	5.36 ± 0.79 ^e^	19.4 ± 2.75 ^c^	6.4 ± 2.07 ^c^	70 ± 3.19 ^b^
Mazzancoio	12.5 ± 0.25 ^b^	11.89 ± 0.39 ^a^	34.1 ± 1.94 ^a^	10.7 ± 0.88 ^a^	8 ± 0.52 ^d^
Rosia	14.5 ± 1.40 ^a^	7.54 ± 0.36 ^d^	33.3 ± 2.79 ^a^	11.1 ± 0.55 ^a^	34 ± 1.71 ^c^
**Location**	***	ns	**	**	ns
Rombiolo	11.9 ± 1.74 ^a^	8.48 ± 1.94 ^a^	21.5 ± 8.85 ^a^	7.2 ± 3.07 ^a^	73 ± 35.08 ^a^
Maierato	11.1 ± 2.16 ^b^	8.33 ± 2.02 ^a^	20.4 ± 9.32 ^b^	6.7 ± 3.05 ^b^	73 ± 35.07 ^a^
**Variety**					
Modern	10.5 ± 1.69	8.52 ± 0.97	15 ± 4.55	5.1 ± 1.86	99 ± 1.00
Old	11.6 ± 0.78	5.36 ± 0.79	19.4 ± 2.75	6.4 ± 2.07	70 ± 3.00
Landrace	13.5 ± 1.45	9.71 ± 2.25	33.7 ± 2.38	10.9 ± 0.75	21 ± 13.00

FP, flour protein (% d.m.); YI, yellow index (*b**); WGC, wet gluten content (%); DGC, dry gluten content (%); GI, gluten index (%). Mean values ± standard deviation are reported for each variable. Different letters in the same column indicate significant differences according to Tukey HSD test at *p* ≤ 0.05, while ***, **, ns indicate the following levels of significance for each ANOVA model: 0.001, 0.01, not significant. Only the mean values ± standard deviation are shown for the variety component for descriptive purposes. Sources of variability are indicated in bold in the first column of the table.

**Table 4 foods-11-02359-t004:** Alveograph and farinograph results.

	DEW	ARPL	FDDT	FS	FSD	WAC	FQN
**Genotype**	***	***	***	***	***	***	***
Altamira	120.75 ± 26.67 ^bc^	1.43 ± 0.39 ^bc^	1.83 ± 0.32 ^bcd^	1.88 ± 0.69 ^c^	105 ± 13 ^c^	51.7 ± 3.13 ^c^	22.0 ± 4.60 ^d^
Bologna	260.80 ± 57.63 ^a^	1.46 ± 0.50 ^b^	1.75 ± 0.49 ^cd^	14.80 ± 3.76 ^a^	54 ± 13 ^e^	55.6 ± 2.08 ^b^	47.8 ± 9.50 ^a^
Solehio	99.75 ± 29.55 ^cd^	1.39 ± 0.17 ^bc^	1.95 ± 0.45 ^bc^	1.83 ± 0.93 ^c^	113 ± 17 ^c^	50.3 ± 1.55 ^cd^	27.8 ± 8.88 ^c^
PR22R58	128.50 ± 53.93 ^bc^	1.13 ± 0.32 ^c^	1.53 ± 0.41 ^d^	1.65 ± 0.71 ^c^	101 ± 29 ^c^	51.1 ± 1.69 ^c^	23.8 ± 6.65 ^c^
Abbondanza	145.50 ± 6.06 ^b^	0.49 ± 0.21 ^d^	2.13 ± 0.14 ^ab^	3.23 ± 1.38 ^b^	85 ± 9 ^d^	48.4 ± 1.02 ^d^	38.2 ± 13.13 ^b^
Mazzancoio	30.91 ± 3.88 ^e^	2.30 ± 0.11 ^a^	1.91 ± 0.11 ^bc^	0.54 ± 0.05 ^d^	177 ± 5 ^a^	66.5 ± 3.34 ^a^	20.7 ± 1.59 ^e^
Rosia	81.25 ±15.21 ^d^	0.35 ± 0.06 ^d^	2.35 ± 0.12 ^a^	1.83 ± 0.34 ^c^	145 ± 14 ^b^	56.6 ± 1.91 ^b^	34.3 ± 1.89 ^c^
**Location**	**	**	ns	***	***	ns	***
Rombiolo	135.46 ± 73.16 ^a^	1.14 ± 0.66 ^b^	1.95 ± 0.34 ^a^	2.13 ± 1.16 ^a^	121 ± 38 ^a^	54.3 ± 5.18 ^a^	34.1 ± 13.77 ^a^
Maierato	112.38 ± 73.57 ^b^	1.31 ± 0.69 ^a^	1.89 ± 0.46 ^a^	1.41 ± 0.74 ^b^	102 ± 40 ^b^	54.3 ± 6.96 ^a^	27.2 ± 8.45 ^b^
**Variety**							
Modern	152 ± 77.1	1.35 ± 0.38	1.76 ± 0.44	5.04 ± 6.02	93 ± 30	52.2 ± 2.98	30.3 ± 12.79
Old	145 ± 6.06	0.49 ± 0.21	2.13 ± 0.14	3.23 ± 1.38	85 ± 9	48.4 ± 1.02	38.2 ± 13.13
Landrace	56.1 ± 27.9	1.32 ± 1	2.13 ± 0.25	1.18 ± 0.7	161 ± 19	61.6 ± 5.73	27.5 ± 7.14

DEW, deformation energy W (×10^−4^ J); ARPL, alveograph ratio (P/L); FDDT, farinograph dough development time (min); FS, farinograph stability (min); FSD, farinograph softening degree (BU); WAC, water absorption capacity 500 BU (%); FQN, farinograph quality number (mm). Mean values ± standard deviation are reported for each variable. Different letters in the same column indicate significant differences according to Tukey HSD test at *p* ≤ 0.05, while ***, **, ns indicate the following levels of significance for each ANOVA model: 0.001, 0.01, not significant. Only the mean values ± standard deviation are shown for the variety component for descriptive purposes. Sources of variability are indicated in bold in the first column of the table.

**Table 5 foods-11-02359-t005:** Quality characteristics of bread obtained from the different genotypes.

	LBV	BH	BW	PO	CT
**Genotype**	***	***	***	***	***
Altamira	436.3 ± 53.9 ^b^	79.5 ± 6.07 ^bc^	135.9 ± 4.40 ^c^	7.0 ± 0.06 ^a^	1.0 ± 0.00 ^a^
Bologna	442.5 ± 44.9 ^ab^	81.7 ± 5.68 ^b^	139.7 ± 4.32 ^bc^	6.0 ± 0.00 ^b^	1.0 ± 0.00 ^a^
Solehio	385.0 ± 22.7 ^c^	75.5 ± 3.35 ^c^	136.7 ± 3.26 ^c^	7.0 ± 0.06 ^a^	1.0 ± 0.00 ^a^
PR22R58	406.3 ± 54.1 ^bc^	78.9 ± 7.32 ^bc^	137.0 ± 2.51 ^c^	7.0 ± 0.00 ^a^	1.0 ± 0.00 ^a^
Abbondanza	485.0 ± 58.2 ^a^	88.3 ± 4.35 ^a^	130.9 ± 2.54 ^d^	6.0 ± 0.05 ^b^	2.0 ± 0.00 ^b^
Mazzancoio	385.6 ± 18.9 ^c^	58.1 ± 2.57 ^d^	150.8 ± 6.84 ^a^	6.0 ± 0.05 ^b^	2.0 ± 0.00 ^b^
Rosia	440.0 ± 45.7 ^b^	79.9 ± 6.73 ^bc^	143.7 ± 3.49 ^b^	6.0 ± 0.00 ^b^	2.0 ± 0.00 ^b^
**Location**	***	***	***	ns	ns
Rombiolo	448.7 ± 58.1 ^a^	80.0 ± 10.91 ^a^	137.7 ± 5.66 ^b^	6.4 ± 0.50 ^a^	1.4 ± 0.50 ^a^
Maierato	402.9 ± 40.3 ^b^	74.9 ± 8.77 ^b^	140.8 ± 8.17 ^a^	6.4 ± 0.50 ^a^	1.4 ± 0.50 ^a^
**Variety**					
Modern	418 ± 50.1	78.9 ± 6.02	137 ± 3.87	6.8 ± 0.44	1.0 ± 0.00
Old	485 ± 58.2	88.3 ± 4.35	130.9 ± 2.54	6.0 ± 0.05	2.0 ± 0.00
Landrace	413 ± 44.1	69 ± 12.21	147 ± 6.42	6.0 ± 0.04	2.0 ± 0.00

LBV, loaf bread volume (cm^3^); BH, bread height (mm); BW, bread weight (g); PO, porosity; CT, crust texture. Mean values ± standard deviation are reported for each variable. Different letters in the same column indicate significant differences according to Tukey HSD test at *p* ≤ 0.05, while ***, ns indicate the following levels of significance for each ANOVA model: 0.001, not significant. Only the mean values ± standard deviation are shown for the variety component for descriptive purposes. Sources of variability are indicated in bold in the first column of the table.

## Data Availability

The data that supported the results of this study are available from the corresponding author upon request.

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
