# Peer review of "A Comparative Assessment of Agronomic and Baking Qualities of Modern/Old Varieties and Landraces of Wheat Grown in Calabria (Italy)"

_foods, 2022, doi:10.3390/foods11152359_

Round 1

Reviewer 1 Report

This manuscript presents a comparative assessment of agronomic and baking qualities of seven genotypes of wheat (modern/old varieties and landraces). The review was carried out thoroughly and in detail and is very interesting. This work presents a complete review the comparative characteristics of different varieties which gives a greater view on the rheological properties of the dough obtained from tchem and also quality of bread. The manuscript is very concrete and with very completes information. Such study is important in the field of cereal science.

I have a few comments regarding the formal requirements of the manuscript Following issues/points should be taken into considerations:

Please note the appropriate line spacing in the manuscript. First, a smaller space between lines was used, and then a larger space. Please check it and correct it according to the journal's requirements (see pages 7, 8, 9, 10, 12, 13, 14, 15, 16).

LINE 401/p. 12 Remove the space before the percent sign 9.0_%

Please do the same on LINES: 442, 443, 444

LINE 162,163 ” Color  parameters of the flour, in the color space, L*, a*, b*, were determined by Chromameter CR-300 (Minolta, Osaka, Japan), under the illuminant, D65. This information is repeated further in subsection 2.6 Bread-making and bread quality evaluation.

I recommend that this information be removed from this subchapter and supplemented in subchapter 2.6

TABLE 2, 3, 4, 5

The values of the standard deviation should be presented to the hundredths (0.15). Please include this in the tables 2, 3, 4, 5. Please also remove the space between the sign and after the sign” ±

Reviewer 2 Report

Dear authors,

After reading the manuscript :” A comparative assessment of agronomic and baking qualities of modern/ old varieties and landraces of wheat”, I believe that this manuscript showed some parts of scientific necessity, especially in the field of biodiversity protection of grains and value added local bakery products production.  In view of the above I recommend to give the article to the authors for correction, and below I have written some suggestions in an attempt to improve it.

L2:  “grown in Calabria” should be added in the title to accurately describe the genotype of wheat study in this manuscript.

L25-26: The statement “However, the mixed use…and landraces could provide an opportunity for…” is not appropriate in abstract, for it’s only a speculation without any provement in this manuscript.

L46, L285 and L449: “W” should be change to be “DEW”, and definite it at the first time.

L58: It was stated that “Old wheat varieties include landraces”. If so, the title, data analysis and the discussion should be changed accordingly.

L138: Here, HP were used for the abbreviation of the plant earing period and we can find data of HP in table 2. But in L214, there is no note for HP, only heading period for HD. In the results and discussion, both earing period and heading period were used. Though they were the same actualy, these parameters should be used consistently throughout the manuscript. Similar situation with degree of farinograph softening degree and farinograph dough development time. When abbreviation was use, the full name should be shown at the same time for the first time, and then abbreviation should be used instead of full name in the following sentences. Both avvreviation and full name were used for FDDT, FSD, WAC, et al..

L172: Receipt and process paramenter such as proofing time, baking time should be illustrated clearly in method 2.6, because the water amount, mixing time, proofing time, et al. all have great influence on the final bread quality.

L248: What What does “value” refer to? The statement in the manuscript should be more accurite.

L271& 275: ARPL should be used instead of P/L.

L233-L234: “…by high hardness…” should be “by higher hardness”; “…was detected for modern…” should be “was detected among modern…”. In some places, the statements are lengthiness, made it hard to understand. So, this manuscript needs language corrections to make statements more brief and concise, and I recommend checking it throughly by a native speaker.

L292: Data of crumb and crust characteristics should be added to table 5, because the importance of them are equal to loaf volue.

L339: This paragraph should be divided into three paragraphes, before “Considering that….” and “Taking into consideration…”, to make it more clearly.

L342-L346: There is no need to illustrate the definition of the hectolitre weight, unless you can relate it to qualities of flour, dough or bread.

L358: “Rosia, due to low yield,…”. Its not reasonable to state that the highest protein content was the result of low yield, even though they have analysis the correlation between grain yield and protein content later.

L358: “a difference greater than 1.5% was detect…”. When the result of comparison was stated, it should be based on the significance shown after analysis rather than the data themselves.

L393-395: There is an error in the sentence order, which have led to a logical error.

L462: What What does “they” and “this variabilityrefer to? The statement in the manuscript should be more accurite.
